# Enhancing Cognitive Function in Older Adults through Processing Speed Training: Implications for Cognitive Health Awareness

**DOI:** 10.3390/healthcare12050532

**Published:** 2024-02-23

**Authors:** Pai-Lin Lee, Chih-Kun Huang, Yi-Yi Chen, Hui-Hsiang Chang, Chun-Hua Cheng, Yu-Chih Lin, Chia-Li Lin

**Affiliations:** 1Graduate School of Adult Education, National Kaohsiung Normal University, Kaohsiung 80201, Taiwan; 2Department of Social Work, National Quemoy University, Kinmen 892009, Taiwan; simonhuang@nqu.edu.tw; 3Wenzao Chinese Language Center, Wenzao Ursuline University of Languages, Kaohsiung 807679, Taiwan; c09901@ms.wzu.edu.tw; 4Center for English Language Teaching, Wenzao Ursuline University of Languages, Kaohsiung 807679, Taiwan; 99125@mail.wzu.edu.tw; 5Occupational Therapy Department, Kaohsiung Municipal Kai-Syuan Psychiatric Hospital, Kaohsiung 802511, Taiwan; 810854003@mail.nknu.edu.tw; 6Division of General Internal Medicine, Department of Internal Medicine, Kaohsiung Medical University Hospital, Kaohsiung Medical University, Kaohsiung 807377, Taiwan; springfred@kmu.edu.tw; 7Department of International Business, Ming Chuan University, Taipei 111, Taiwan; linchiali0704@yahoo.com.tw

**Keywords:** healthy aging, health literacy, cognitive training, cognitive function, brain processing

## Abstract

It may be possible to enhance adults’ cognitive health and promote healthy aging through processing speed training using the Useful Field of View (UFOV) related activities and software. This study investigated the impact of utilizing UFOV on processing speed improvement in older adults in response to the growing global attention on cognitive health and aging issues. In this quasi-experimental study, 22 individuals (mean age ± SD = 71.9 ± 4.8) participated in the experimental group, and 20 community-based participants (mean age ± SD = 67.1 ± 4.8) were in the control group. The intervention involved ten sessions of UFOV training, each lasting 60 min, conducted twice a week for the experimental group while the control group engaged in volunteer service activities. Measurements of Counting Back, Fabrica, Double-Decision, and Hawkeye were administered to all participants before and after the intervention. The results showed significant improvements in the experimental group for the four measurements (*p* ≤ 0.01, 0.05, 0.001, 0.001) and non-significant gains in the control group (*p* ≥ 0.05) for all. Furthermore, mixed repeated-measures ANOVA analysis, with time 1 pre-test measures as the covariate, revealed significant interaction effects between time and group for all measurement indicators (*p* = 0.05, 0.01, 0.05) except for Fabrica (*p* > 0.05). In conclusion, these findings support the effectiveness of UFOV cognitive training interventions in enhancing specific cognitive abilities.

## 1. Introduction

Age-related cognitive decline is prevalent in older adults [1,2]. This includes deficits in memory, attention, executive function, and processing speed [3,4,5,6,7,8]. This processing speed theory centers on the idea that aging is associated with a decrease in the speed of processing operations, which in turn leads to impairments in cognitive functioning [1,2]. There are various types of non-pharmacological cognitive interventions for cognition improvement that are available. Cognitive training for maintaining cognitive health is gaining attention, though its efficacy and evidence remain controversial or limited [9,10,11,12]. Wang et al. [13] conducted a meta-analysis and found that game-based brain training helped improve processing speed, selective attention, and short-term memory in older adults. Ball, Owsley, Sloane, Roenker, and Bruni [14] proposed the effectiveness of a Useful Field of View (UFOV) computer program as a measurement for examining cognitive status. UFOV performance depends on the integrity of visual sensory information and has a strong correlation with individual processing speed [15,16].

A recent neuroimaging study indicated that older adults showed greater improvement in working memory (WM) than younger adults from cognitive training. A study utilizing 3T MRI scanning showed that computerized multi-domain cognitive training (MDCT) may protect patients with amnestic mild cognitive impairment (MCI) against gray matter (GM) volume loss and showed the potential of MDCT in preserving general cognition [17]. The scaffolding theory of aging and cognition (STAC) provides a model for the etiology of age-related compensatory neural activity. In this model, scaffolding (compensatory neural activity) is defined as the recruitment of additional circuitry to support declining brain function that has become noisy, inefficient, or both [18]. In addition, a systematic review of 38 studies published between 1984 and 2011 suggested that pre–post training effect sizes for intervention groups ranged from 0.06 to 6.32 for traditional cognitive training interventions, 0.19 to 7.14 for neuropsychological software interventions, and 0.09 to 1.70 for video game interventions [19]. Accordingly, computerized cognitive training programs deserve further exploration due to their convenience, adaptiveness, effectiveness, and less labor-intensive advantage.

### 1.1. Processing Speed UFOV Cognitive Training

Recent studies showed that UFOV, as a process-based perceptual cognitive training technique, improves neural connectivity, processing speed, and attention in the elderly [20]. Fausto et al. [21] indicated that cognitive training such as UFOV could improve driving performance and reduce car accidents. McCloskey and Webb [22] revealed that UFOV could lead to safer driving in older adults by improving the speed of processing, reducing divided attention, and quickening reaction time. Some researchers assert that process-based cognitive training tends to produce greater effects and might be easier to transfer than other techniques [9,23]. A randomized controlled clinical trial, the Advanced Cognitive Training for Independent and Vital Elderly (ACTIVE) study, also indicated that UFOV training leads to continuous improvement in the proximal end, with significant effects after ten years [1,24]. In Rebok’s research [1], the measures of processing speed involved three UFOV tasks, which required the location and identification of information with 75% accuracy under different cognitive demand levels. With a medium-to-large effect size of 0.66, this improvement lasted for ten years, and 70.7% of the speed-trained participants achieved or exceeded their baseline cognitive ability, with 48.8% (*p* < 0.01) of the control group participants achieving it. UFOV training can maintain health and well-being through various methods [20,25,26,27]. More importantly, UFOV training may delay the onset of dementia in the long term [28]. In summary, cognitive visual processing speed ability seems closely related to elderly physical and mental health, mobility safety, daily life ability, and driving safety. Importantly, this ability could be enhanced through the practice of UFOV.

### 1.2. Purpose of the Study

Studies have proven that many elderly people often encounter difficulties in their daily life due to poor visual functioning, resulting in accidents when driving and the inability to search for objects quickly or move freely [29]. Therefore, this study employs UFOV video games and traditional processing speed activities designed by the authors as a set of UFOV training courses for the elderly to understand its effect on effective vision and cognitive abilities. The aim of this study is to investigate whether structured UFOV training courses promote the cognitive abilities of the elderly in Taiwan. Previous studies, as mentioned above, are predominantly oriented towards Western cultures, leaving uncertainty about how their findings would translate when applied to Eastern cultures. Due to the general unfamiliarity among older people in using cognitive games for brain health, our UFOV training introduces a novel approach to engaging this demographic. This method not only aims to enhance their cognitive abilities but also serves as an innovative form of public health education, thereby increasing health awareness among the elderly. Furthermore, our study is unique as it is the first to implement UFOV gaming in Taiwan, an Eastern country. We aim to assess its beneficial effects on the cognitive health of elderly residents within Taiwanese communities. Community residents from a metropolitan area in southern Taiwan, who were older, were recruited as participants from March 2019 to April 2019. The training and testing of cognitive processing speed ability aimed to improve the cognitive function of these older adults. The researchers intended to identify differences both between and within the experimental and control groups. 

## 2. Materials and Methods

### 2.1. Participants

The rights and interests of research participants and the research process were reviewed by the National Cheng Kung University Human Research Ethics Committee (NCKU HREC-E-110-190-2, approved on 26, 02, 2019). The participants were older adults in Taiwan who were recruited from the community. The recruitment procedure was initiated through the dissemination of informational posters. Interested individuals were invited to register for our program, with eligibility criteria stipulating the exclusion of participants diagnosed with depression or psychotic disorders. All applicants who registered for the study met the qualifications and were admitted, and their basic information is as shown in Table 1.

Forty-two healthy adults participated in the study with a mean age of 71.9 ± 4.8 (range = 65–81) years for the experimental group and 67.1 ± 4.8 (range = 59–75) years for the control group. Participants were recruited by posters and all participants were fluent in Mandarin Chinese, negating any communication issues. The mean Montreal Cognitive Assessment Test score [30] of the experimental group was 26.4 ± 2.6, and that of the control group was 25.4 ± 3.8. All participants signed an informed consent form to participate in the study, and the study was approved by the local ethics committee (case number: 107-2410-H-017-008-MY2). This course did not charge any fees to ensure that participation motivation was not affected by monetary issues. The social–demographic data of the participants are shown in Table 1.

### 2.2. Pre and Post Test Measures

Four cognitive function measures were administered to the participants before and after the intervention: Counting back, Fabrica, Double Decision, and Hawk Eye. All are measured by response time (second), and accordingly, shorter response times indicate better performance. The assessment tools were administered at two distinct time points: the pre-test was conducted within the first week of initiating the training course, and the post-test was completed within a week following the conclusion of the 10-week course.

(1)Counting Back

Counting back is a test of attention, working memory, and response speed. Participants were told to count backwards from 100 to 50. The task completion time was recorded by a stopwatch in seconds. The time to task completion shows the participant’s reaction speed. The counting back measure has been used in other studies [31,32] for attention and executive functions, which are related to processing speed. 

(2)Flash Fabrica

The Flash Fabrica number recall game is an online instant memory game that tests concentration, reaction, memory, and effective vision. There are ten numbers from 0 to 9 which appear randomly and are also randomly scattered around the screen in each instance. The numbers are obscured after about 0.7 s. Participants must recall the location of the hidden numbers and click them from small to large. The total number of digits appearing in each successive round is determined by the number of correct answers previously received. The higher the correctness rate, the more numbers appear in the next question, and so, difficulty is adjusted based on performance. After completing a session of 10 instances, the computer will calculate the “brain age” score. Slow reaction speeds and incorrect clicks lead to a higher brain age score, with the optimal score being 20. This score indicates a brain age equivalent to that of a 20-year-old. Nouchi [33] claimed that combining the Brain Age training game with others improved executive function and processing speed [33].

(3)Double Decision

This game from the Posit Science Company was designed to train effective field of vision, accelerate processing speed, and improve concentration. A vehicle and a road sign first appear on the screen, and then disappear after a few seconds. Then, two different types of cars appear on the screen afterward. Participants must respond quickly to distinguish which vehicle is present on the screen, choose the correct one, and then select the direction of the road sign (e.g., Route 66). With additional correct answers, the objects disappear faster. Milliseconds are used as the result of the game. This game’s method of double decision training has shown improved speed of processing performance versus baseline to be lasting to 2 years [34] and 10 years [1] and is even associated with a wide array of cortical regions that provide unique contributions to performance [35]. 

(4)Hawk Eye

Posit Science’s Brain HQ features the Hawk Eye game, aimed at improving visual precision. This game challenges players to quickly identify a specific bird within a fleeting view of a bird flock. The task emphasizes detecting birds in peripheral vision during brief on-screen appearances. Difficulty increases as players advance, with birds in the exercises resembling each other more closely, being positioned further apart, and set against increasingly intricate backgrounds [36]. This progression is designed to steadily enhance the user’s visual attention and detail recognition skills. Scoring is based on the player’s ability to correctly locate the target bird. Participants’ scores are measured in milliseconds. As they improve, the birds flash for fewer milliseconds, giving them a lower (better) score. This game was also referenced elsewhere as improving cognitive functions [37,38].

### 2.3. Intervention Training Tasks

Older adults residing in Southern Taiwan participated in a five-week trained course, held twice a week for 60 min each session, aimed at enhancing the cognitive abilities of the elderly, particularly focusing on expanding effective vision, strengthening concentration, and enhancing processing speed.

The intervention involved participants playing traditional non-online activities, as well as online Useful Field of View (UFOV) games, which were provided by the Posit Science company. The experimental group underwent training that was divided into two parts per session. First, participants engaged in 30 min of traditional processing speed-related activities requiring visual attention and quick reactions. For example, in one session, participants were given 2-4 flags of red vs. green and black vs. white, and then, one of the four colors appeared randomly on a slide. When the participants saw a red-colored slide, they needed to wave the green flag, and when they saw white, they had to wave the black flag. After the traditional intervention activity, participants engaged in 30 min of UFOV video game play, which was intended to improve the cognitive processing speed of older adults. The reason for dividing the intervention into two parts was to preserve eye health, thus limiting the visual game playing time. 

Training session durations in various studies show considerable variation, ranging from as short as one day to programs extending over 10 weeks or more. These studies include a diverse array of schedules, such as five sessions spread across two to five weeks, 10 sessions over five to six weeks, and 16 sessions over eight weeks [3]. Penner et al. [39] found that distributed training using their BrainStim tool was more effective for cognitive enhancement than high-intensity training or no training. Lampit, Hallock, and Valenzuela [40] conducted a meta-analysis to determine the efficacy of computerized cognitive training (CCT) in older adults. The study concluded that CCT is less effective with more than three sessions per week or sessions under 30 min. Accordingly, the length and number of sessions are appropriate in our study.

The active control group comprised a convenience sample, selected for the pragmatic ease of data acquisition. Activities undertaken by this group adhered to their routine and institutional schedules. These group members participated in a distinct set of activities, notably encompassing 300 min dedicated to Buddhist Studies and an equal duration committed to volunteer services within temple environments or broader community settings. Comprehensive schedules delineating the interventions for both the experimental and control cohorts are outlined in Table 2 and Table 3, thereby elucidating the specific elements integral to each group’s intervention methodology. Both the experimental and control group met in groups for the training course.

### 2.4. Statistical Analysis

The demographic characteristics, e.g., age and education level, were analyzed by a *t*-test, and gender was analyzed by the chi-square method. The effect size (Cohen’s d) of both groups was also calculated in order to provide information about the magnitude of the intervention effects. The mixed-design ANOVA compensates for the inherent violation of independence in repeated measures by accounting for observation independence [41]. Additionally, it is robust to normality violations, especially with similarly sized sample groups. In practical research, where large samples are often unfeasible, using small samples can be justified for exploratory studies like ours.

## 3. Results

In evaluating normality, skewness for a normal distribution ideally approximates zero. For the experimental group, skewness values at pre-test were 0.97, 1.24, 0.27, and 3.62 for Counting back, Fabrica, Double Decision, and Hawkeye, respectively, shifting to 1.33, 2.66, 2.70, and 0.99 at post-test. Kurtosis values were −0.15, 3.78, −1.34, and 14.60 at pre-test, changing to 1.62, 10.14, 8.46, and −0.23 at post-test. The control group presented pre-test skewness of 1.11, 2.24, 1.30, and 0.38, and post-test values of 3.12, 2.03, 1.88, and 1.91, with kurtosis values at 1.06, 7.50, 0.23, and −0.93, and 11.17, 7.67, 2.94, and 3.21, respectively, at pre-test and post-test. These significant deviations from zero suggest non-normality, recommending the use of non-parametric analysis [42].

In this study, the Wilcoxon Signed-Rank Test revealed significant intragroup differences in cognitive scores pre- and post-intervention within the experimental group. Specifically, significant median gains were observed in the ‘Counting back’ (Z = −2.869, *p* = 0.004), ‘Fabrica’ (Z = −2.138, *p* = 0.033), ‘Double Decision’ (Z = −4.075, *p* < 0.001), and ‘Hawkeye’ (Z = −3.251, *p* = 0.001) tasks. Conversely, the control group did not demonstrate significant improvements, with all tasks showing non-significant changes (all *p* > 0.05). These findings suggest that the experimental intervention had a measurable impact on cognitive performance in the experimental group, as opposed to the control group, as shown in Table 4.

In addition, the effect size measured by Cohen’s d suggested that the Double Decision task in the experimental group achieved a large intervention effect (effect size = 1.379), while the Hawkeye task reached a moderate intervention effect (effect size = 0.791). Moreover, the Counting back and Fabrica tasks in the experimental group showed small intervention effects, with effect sizes of 0.391 and 0.406, respectively. The training effects also indicated that 77%, 68%, 95%, and 100% of the experimental participants, compared to 20%, 35%, 75%, and 55% of the control group, performed at or above their baseline level in Counting back, Fabrica, Double Decision, and Hawkeye tasks, respectively. 

According to Table 5, the mixed-design ANOVA revealed an interaction between the time factor (within subjects) and the group factor (between subjects) for the dependent variables of Counting back, Double Decision, and Hawkeye. This indicated that the intervention group significantly improved their task performance in the three dependent variables from time 1 to time 2 when compared with the control. However, no interaction effect was found for Fabrica.

## 4. Discussion

Taiwan is among one of the fastest aging societies in the world, and the results of the study highlight its significance. Based on the results of our UFOV research, we conducted the following discussions on both within-group and between-group differences.

The results showed that UFOV training programs can improve visual processing speed. In the four cognitive tasks tested, the experimental group’s answer accuracy and reaction speed improved significantly on all the four tests, including Counting back, Fabrica, Double Decision, and Hawkeye games, which generally represent attention and processing speed; meanwhile, the control group did not show any improvements. Our findings are consistent with previous studies [1,20,24].

The reasons for our results might be: (a) Real-world Transfer and Cognitive Enhancement: UFOV consistently demonstrates real-world transfer benefits over 3–10 years [43,44], suggesting improved cognitive functions, including processing speed. (b) Enhanced Neural Efficiency: UFOV is associated with reduced neural effort during task completion, as seen in the decreased neural activation in regions linked to cognitive control and attention [45,46]. This reflects improved cognitive processing speed. (c) Increased Connectivity and Efficiency: UFOV enhances resting-state functional connectivity in areas involved in task performance [47], indicating a more efficient utilization of neural resources for improved processing speed. (d) According to Hebbian learning theory, any two neurons or a group of neurons that are active at the same time will form a stronger connection between them. Consequently, activity in one will facilitate the other [48].

In addition, the Scaffolding Theory of Aging and Cognition (STAC) theory claims that cognitive training can provide scaffolding effects, strengthen cognitive functions, and enable participants to benefit from cognitive interventions [49]. The current study using a UFOV training program involves concentration, attention, and processing speed, so participants naturally made significant progress through training.

The outcomes of our study demonstrate that there were notable distinctions between the experimental group and the control group regarding their performance in the Counting back, Double Decision, and Hawkeye tasks. This signifies that individuals in the experimental group exhibited a significantly different level of performance compared to those in the control group. It indicates that the cognitive training program had a measurable impact on their task performance.

The individuals in the experimental group experienced substantial benefits as a result of participating in the cognitive training program. This suggests that the UFOV training program had a positive and significant effect on the cognitive abilities of the participants, consistent with previous studies [20,22], leading to improved performance in the mentioned tasks.

In addition, these positive effects can be attributed to the effectiveness of the UFOV continuing training courses. These courses are designed to enhance concentration, reaction time, and processing speed, which, in turn, contributed to the significant benefits observed. Furthermore, factors discussed in a prior section concerning differences within the experimental group may have also played a role in these overall improvements.

An interaction (time × group) effect was found in all the measures except the Fabrica. This may be due to the Fabrica task needing more training sessions for the positive effect to be seen. Even though the control group participated in various life-enriching activities such as religious worship, meditation, music appreciation, culinary activities, and community service, there was no significant effect on cognitive function. This might suggest that these activities need to be well organized and designed in order to gain the benefits. Precise targeting is important in that cognitive training should target specific cognitive mechanisms, such as executive attention–control processes or core executive functions, rather than broad goals [50]. The key differences between the Flash Fabrica and the other UFOV two games (Double Decision and Hawk Eye) are that Flash Fabrica entails the following: (a) the game primarily targets instant memory recall and concentration, requiring players to remember and locate numbers; (b) players recall the location of briefly shown numbers and click them in order, with the game increasing the number of digits based on performance; (c) the game uses a unique “brain age” score based on reaction speeds and accuracy, with a lower score indicating better performance. Accordingly, it seems reasonable that no interaction effect was found in the study.

Some studies suggest that the time that the elderly spend playing video games should not be too long; otherwise, it might damage their eyesight [51]. In order to prevent eyesight problems, the time for video games was limited to 30 min (including the lecture and discussion for about 10 min). Special attention should be paid to the internet network environment since these are online games. Because of internet difficulties, network technicians were required to resolve this issue before the sessions.

(1)Strengths

Overall, this study demonstrates the significant benefits of a well-designed cognitive training program, particularly in an aging society like Taiwan, and adds valuable knowledge to the field of cognitive enhancement through targeted training interventions. The strengths of the study include: 1. Effective UFOV Training Program: The study highlights the effectiveness of the UFOV training program in improving visual processing speed. Participants in the experimental group showed significant improvements in accuracy and reaction speed across four cognitive tasks, indicating a substantial benefit of the training. 2. Notable Intergroup Difference: There were significant differences between the experimental and control groups in tasks like Counting back, Double Decision, and Hawkeye, indicating the effectiveness of the UFOV training program in enhancing cognitive abilities. 3. Importance of Precise Targeting in Cognitive Training: This study underscores the necessity of targeting specific cognitive mechanisms for effective cognitive training, as broad approaches may not yield significant benefits. 4. Consistency with Prior Research: Our findings align with previous studies, as mentioned above, thereby reinforcing the validity and reliability of our results

(2)Study limitations

Lack of Randomization and Blinding. This study employed the Useful Field of View (UFOV) technique for cognitive performance assessment. However, it is important to note that our methodology did not include randomization or blinding. Randomization: The absence of randomization in our study design means that potential baseline differences between participant groups might have influenced the results, introducing a risk of selection bias. Blinding: Without blinding, there is a possibility of performance and detection biases, as participants’ and researchers’ behaviors could be influenced by their awareness of the intervention. Implications: Due to these limitations, the findings should be interpreted with caution. They may reflect the biases introduced by the study design rather than the effectiveness of the UFOV technique itself. We suggest considering these results as preliminary. In addition, although the measurements in the study require individual attention to test the immediate response and have been used elsewhere, we acknowledge the absence of information on reliability and validity, which constitutes a limitation of the study.

Significant age differences exist between the two groups, which might influence the results. Therefore, these differences should be carefully considered for future study. Though we propose several theories to interpret the results, we acknowledge that our study does not clearly identify which specific reason is actually responsible for these outcomes.

(3)Implications and future studies

The findings from this study have significant implications for elderly populations in Taiwan and elsewhere. The effectiveness of the UFOV training program in improving visual processing speed, accuracy, and reaction time in cognitive tasks highlights its potential as a valuable tool for cognitive enhancement in aging societies. This program can help mitigate age-related declines in cognitive functions like attention and processing speed, enhancing overall quality of life. 

Our study represents a groundbreaking addition to the existing body of research on cognitive enhancement through UFOV (Useful Field of View) training. While previous studies have established the effectiveness of UFOV training in improving cognitive functions, our research is pioneering in its application in Taiwan, a culturally distinct context. This is significant because it demonstrates the cross-cultural applicability and relevance of UFOV training, underscoring its potential as a universal tool for cognitive improvement.

As mentioned earlier, our study’s intervention schedule of 60 min twice a week for five weeks aligns with recommendations from previous research. Nonetheless, further investigation is needed to determine the longevity of these effects and the extent of their transferability to other cognitive domains or real-life scenarios.

The study demonstrates enhanced cognitive functions in participants post-training, especially in visual processing, accuracy, and reaction time, due to UFOV training. This suggests the transferability of these skills to daily activities, potentially improving decision-making and attention. The research also points to the practical application of such training in diverse environments like community centers for the elderly and cognitive rehabilitation programs. However, the duration of these cognitive improvements remains uncertain, highlighting the need for further research to determine the longevity of the training effects.

## 5. Conclusions

This study offers pivotal insights for both health care and older adult education, particularly in societies like Taiwan, which are experiencing rapid aging. The effectiveness of the UFOV training program in improving cognitive abilities among the elderly, specifically in terms of visual processing speed, accuracy, and reaction time, is of significant relevance. These findings are invaluable for health care professionals and educators focused on older adults, providing a potential pathway to mitigate age-related cognitive decline. While the limitations of the study, such as the absence of randomization and blinding, suggest that these results should be interpreted with caution, they nonetheless indicate the promising role of targeted cognitive training in enhancing the cognitive health and overall well-being of the elderly. Thus, this research contributes essential knowledge to both health care and the field of older adult education, underscoring the importance of specialized interventions in these areas.

## Figures and Tables

**Table 1 healthcare-12-00532-t001:** Baseline characteristics of the participants for both groups.

Characteristics	Experimental, *N* = 22 Mean (SD)	Control, *N* = 20 Mean (SD)	*p*
Age	71.9 ± 4.8 (65–81)	67.1 ± 4.8 (59–75)	0.003
Female, n %	19 (86.4)	16 (80.0)	0.580
Education, mean ± SD (range)	8.1 ± 4.4 (0–15)	10.1 ± 3.5 (6–16)	0.126
MOCA *	26.4 ± 2.6	25.4 ± 3.8	0.291

Note. * Montreal Cognitive Assessment, low value indicated low cognitive function.

**Table 2 healthcare-12-00532-t002:** The content and objectives of the intervention schedule for the experimental group.

Session.	Content	Objective
1	A 30 min introduction on the importance of Useful Field of View (UFOV); 30 min practicing video game: Double Decision Identify a vehicle and locate a road sign. The operation of this electronic game involves the appearance of two vehicles and a Route 66 sign on the screen. Participants are required to identify the correct vehicle and the location of the Route 66 sign.	Understanding the importance of UFOV Attention training
2	A 30 min period practicing attentional card game; 30 min practicing “Divided and Selective Attention” UFOV games Identifying different birds and determining their correct positions. The electronic game’s operation entails identifying a uniquely colored bird (black head, darker wings) among a group displayed on screen.	Training: attention and response
3	A 30 min period for nine squares fruit memory game activities; 30 min practicing “Divided and Selective Attention” UFOV games. Identifying different birds and determining their correct positions. This electronic game requires players to spot a distinctively colored bird (with a black head and darker wings) among a group on the screen, as per the aforementioned description.	Training: attention and response
4	A 30 min period for advance nine squares game; 30 min practicing “Divided and Selective Attention” UFOV games From session four to the final session, for participants engaged in two tasks: a) identifying birds transitioning from uniform, simple colors to complex, multilayered hues and detailed features (like stripes), necessitating enhanced concentration for subtle distinctions; b) observing a color progression where the initial stark contrast between subject and target gradually narrowed, increasing the challenge until the colors closely resembled each other. Background settings varied from beaches and mountains to general scenic landscapes.	Training: attention and response
5	A 30 min dragon–tiger divided attention game: practicing “Divided and Selective Attention”; 30 min practicing “Divided and Selective Attention” UFOV games. Content is same as session four description.	Attention training-multitasking, attention and response
6	A 30 min object-finding game: finding specific objects in the pictures activity; 30 min practicing “Divided and Selective Attention” UFOV games. Content is same as session four description.	Training: attention and response
7	A 30 min finding differences activity; 30 min practicing “Divided and Selective Attention” UFOV games. Same as session four description.	Training: attention and response
8	A 30 min flag-waving game; 30 min practicing “Divided and Selective Attention” UFOV games. Content is same as session four description.	Processing Speed Training, attention and response
9	A 30 min traditional non-online attention activity; 30 min practicing “Divided and Selective Attention” UFOV games. Same as session four description.	Training: attention and response
10	A 30 min traditional non-online attention activity; 30 min practicing “Divided and Selective Attention” UFOV games. Same as session four description.	Training: attention and response

Note: First 30 min of all sessions above were traditional non-online games, and the next 30 min were online UFOV video games—Double Decision and Hawkeye. Pre-test and post-test measures were given before and after the 10-session intervention.

**Table 3 healthcare-12-00532-t003:** The content and objective schedule for the control group.

Session.	Content	Objective
1	Reading club meeting	Meditation, concentration
2	Cleaning service inside a non-profit organization	Physical activity
3	Reading club meeting	Meditation, concentration
4	Reading club meeting	Meditation, concentration
5	Buddhist Studies	Concentration
6	Practicing African drums	Music appreciation
7	Volunteer Service Activities	Physical activity
8	Supporting community events	Community service
9	Supporting traffic management for the temple activity	Community service
10	Cooking class	Learning cooking

Note. Pre-test and post-test measures were given before and after the 10-session intervention.

**Table 4 healthcare-12-00532-t004:** Difference in the median and mean cognitive scores at pre- and post- timepoints of each group.

	Experimental Median (*N* = 22)			Control Median (*N* = 20)			Experimental	Control
	Pre-	Post-	Md Gain	Pre-	Post-	Md Gain	Intragroup diff Z(*p*)	Intragroup diff Z(*p*)
Counting back	59.50	54.00	−5.50	50.00	50.50	0.50	−2.869 (0.004 **)	−0.112 (0.911)
Fabrica	57.00	51.50	−5.50	52.50	53.00	0.50	−2.138 (0.033 *)	−0.616 (0.538)
Double decision	1653.25	85.25	−1568.00	419.00	290.00	−129.00	−4.075 (0.001 ***)	−1.388 (0.165)
Hawkeye	207.75	61.00	−146.75	65.75	48.75	−17.00	−3.251 (0.001 ***)	−1.445 (0.149)
	Experimental			Control				
Counting back								
M (SD) (sec.)	71.00 (28.49)	60.45 (24.59)		54.45 (23.24)	63.80 (46.58)			
Mean gains	−10.55 (−3.90)			9.35 (23.33)				
Effect size *d*	0.391			−0.23				
At or below baseline, % ^a^	77			20				
Fabrica								
M (SD) (ba)	57.87 (19.63)	50.14 (18.35)		51.67 (19.59)	50.30 (20.07)			
Mean gains	−7.73 (−1.28)			−1.37 (0.48)				
Effect size *d*	0.406			0.069				
At or below baseline, % ^a^	68			35				
Double decision								
M (SD) (ms)	1480.02 (1045.64)	297.25 (445.71)		925.75 (1067.54)	677.85 (867.08)			
Mean gains	−1182.77 (−599.93)			−247.90 (−200.46)				
Effect size *d*	1.379			0.252				
At or below baseline, % ^a^	95			75				
Hawkeye								
M (SD) (ms)	335.93 (453.79)	95.12 (71.93)		66.63 (29.21)	67.68 (51.84)			
Mean gains	−240.81 (−381.86)			1.05 (22.63)				
Effect size *d*	0.791			−0.025				
At or below baseline, % ^a^	100			55				

Notes. *: *p* < 0.05; **: *p* < 0.01; *** *p* < 0.001. Md: median. sec.: second. ms: millisecond. Negative mean gains indicate improvement. ^a^ Calculated as the percentage of participants in each group who were ≥0.66 standard errors of measurement above baseline. ba: brain age, the age range spans from the lowest score of 20 to the highest score of 120.

**Table 5 healthcare-12-00532-t005:** Mixed-design ANOVA results for intergroup differences in each cognitive task.

Source	SS	df	MS	F(*p*)	η_p_^2^
Counting back (Time)	7.49	1	7.49	0.012 (0.901)	0.00
group	913.31	1	913.31	0.59 (0.447)	0.02
Time * group	2073.39	1	2073.39	4.33 (0.044 *)	0.10
Fabrica (Time)	432.65	1	432.65	4.76 * (0.035)	0.11
group	191.49	1	191.49	0.29 (0.594)	0.01
Time * group	211.69	1	211.69	2.33 (0.135)	0.06
Double Decision (Time)	10,721,461.42	1	10,721,461.42	21.62 *** (0.001)	0.35
group	157,992.56	1	157,992.56	0.15 (0.704)	0.00
Time * group	457,8027.23	1	4,578,027.23	9.23 ** (0.004)	0.19
Hawk Eye (Time)	301,106.93	1	301,106.93	5.02 * (0.031)	0.11
group	461,225.10	1	461,225.10	8.77 ** (0.005)	0.18
Time * group	306,371.78	1	306,371.78	5.11 * (0.029)	0.11

Note. *: *p* < 0.05; **: *p* < 0.01; *** *p* < 0.001. Time has 1 = Pre and 2 = Post, and Group has 1 = Experimental and 2 = Control.

## Data Availability

The dataset presented in this research is available with a legitimate request from the corresponding author.

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
