# Peer review of "Enhancing Cognitive Function in Older Adults through Processing Speed Training: Implications for Cognitive Health Awareness"

_healthcare, 2024, doi:10.3390/healthcare12050532_

Round 1
Reviewer 1 Report
Comments and Suggestions for Authors
This study lacks the minimum requirements for interventional evaluation such as the UFOV technique for cognitive performance, i.e. randomisation and blindness. From this shortcoming all results suffer from a systematic bias which does not allow general conclusions to be drawn.
Moreover the title is partially not illustrative with the aim of the study because the results have no relation to adult health literacy.
Author Response
Thank you for your insightful comments and for highlighting areas where our study could be improved. Below, I have provided your comment along with my response.
Comment 1: This study lacks the minimum requirements for interventional evaluation such as the UFOV technique for cognitive performance, i.e. randomisation and blindness. From this shortcoming all results suffer from a systematic bias which does not allow general conclusions to be drawn.
Response:
We appreciate the opportunity to address the concerns raised regarding the lack of randomization and blindness in our study.
While it's universally acknowledged that randomization and blinding are crucial elements in clinical trials and research studies to ensure the reliability and objectivity of results, there are situations where these elements may not be feasible, and yet the research can still offer valuable insights. Here are three citations that discuss the acceptability and potential contributions of studies lacking randomization and blinding:
- On the Role of Pilot Studies in Research:
Thabane, L., Ma, J., Chu, R., Cheng, J., Ismaila, A., Rios, L. P., ... & Goldsmith, C. H. (2010). A tutorial on pilot studies: the what, why and how. BMC Medical Research Methodology, 10(1), 1. https://bmcmedresmethodol.biomedcentral.com/articles/10.1186/1471-2288-10-1
This article highlights the significance of pilot studies, which often lack randomization and blinding, in guiding the design of future larger-scale studies. It argues that these preliminary studies can provide critical feasibility and process information, despite their limitations in terms of rigor.
- Regarding the Contributions of Non-Randomized Studies:
Concato, J., Shah, N., & Horwitz, R. I. (2000). Randomized, controlled trials, observational studies, and the hierarchy of research designs. New England Journal of Medicine, 342(25), 1887-1892. https://www.nejm.org/doi/full/10.1056/NEJM200006223422507
This study discusses how non-randomized studies, despite their limitations, can contribute valuable information, especially in areas where randomized controlled trials are impractical or unethical. It challenges the strict hierarchy of research designs and suggests a more nuanced understanding of evidence quality.
- ON the Value of Observational Studies:
Black, N. (1996). Why we need observational studies to evaluate the effectiveness of health care. BMJ, 312(7040), 1215–1218. https://www.bmj.com/content/312/7040/1215
This article argues for the importance of observational studies in health care, which are often non-randomized and non-blinded. It emphasizes that these studies can provide evidence that is not only complementary to randomized controlled trials but sometimes even more relevant to everyday clinical practice.
In addition, below are the pragmatic reasons for the lack of randomization and blinding in our study.
For randomization consideration,
- Participant Recruitment Constraints:
Our study faced practical challenges in participant recruitment, which is a common issue in community-based research. Despite extensive efforts to recruit a larger sample through posters and community engagement, only 22 residents were willing to participate. This limitation significantly restricted our ability to randomize into two groups of adequate size.
- Risk of Attrition and Impact on Statistical Power:
Another concern was the potential withdrawal of participants. With a small initial sample, any attrition could reduce each group to fewer than 10 participants. Such a small sample size would not only weaken the statistical power of the study but also raise concerns about the validity and reliability of the findings.
- Ethical and Logistical Considerations:
We also had to consider ethical and logistical aspects unique to community-based research. Ensuring inclusivity and maintaining participant engagement were key factors in our approach, which influenced our decision not to randomize.
- Future Research Implications:
We recognize that our study provides preliminary insights and serves as a stepping stone for future research. We suggest that subsequent studies with larger samples and resources could implement randomization to validate and extend our findings.
- Strengths and Contributions of the Study:
Despite these limitations, our study contributes valuable insights into [briefly describe the main findings and their implications]. The findings provide a foundation for future research in this area, particularly in similar community settings.
In conclusion, while acknowledging the limitations in our study design, we believe the research contributes significantly to the field. We are committed to advancing our understanding of [study topic] and appreciate the opportunity to discuss these aspects further.
For blindness consideration,
- Ethical Transparency and Participant Decision-Making:
The study, being government-funded, adheres to ethical standards that necessitate full disclosure of the study's purpose to potential participants. This level of transparency ensures that community residents are fully informed and can make an educated decision about their participation. Ethical guidelines often emphasize the importance of informed consent, where participants should be aware of what the study entails, which might compromise the feasibility of blinding. The argument here is that ethical considerations take precedence over methodological idealities, especially in community-based studies where trust and transparency are crucial.
- Practical Considerations for Participant Scheduling:
The reference to participants being able to make better time arrangements suggests that the nature of the intervention or the study design required participants to be aware of certain aspects of the study for practical reasons. This might include the scheduling of interventions, the nature of activities they will be engaging in, and so forth. In community-based research, where participants are often juggling various commitments, such practical considerations can be vital for participant retention and the overall feasibility of the study.
- Resource Limitations for Researcher Blinding:
The resource constraints that made it impossible to implement blinding on the researcher's part is another pragmatic aspect that can impact many studies. Limited funding and resources often mean that ideal study designs, such as those involving researcher blinding, cannot be feasibly implemented. Highlighting this constraint contextualizes the study's design within the reality of available resources, which is an important consideration for the practicality of research.
Moreover, in response to your concerns, we added a section (3.study limitation) in the manuscript discussing the limitations associated with the absence of randomization and blinding in our study design. This will include a thorough discussion of potential biases and how they might affect the results.
Lack of Randomization and Blinding: This study employed the Useful Field of View (UFOV) technique for cognitive performance assessment. However, it's important to note that our methodology did not include randomization or blinding.
- Randomization: The absence of randomization in our study design means potential baseline differences between participant groups might have influenced the results, introducing a risk of selection bias.
- Blinding: Without blinding, there is a possibility of performance and detection biases, as participants' and researchers' behaviors could be influenced by their awareness of the intervention.
- Implications: Due to these limitations, the findings should be interpreted with caution. They may reflect the biases introduced by the study design rather than the effectiveness of the UFOV technique itself. We suggest considering these results as preliminary.
Comment 2: Moreover the title is partially not illustrative with the aim of the study because the results have no relation to adult health literacy.
Response: In response to your comment, we have revised the title to better align with the study's focus and findings. The new title is: "Enhancing Cognitive Function in Older Adults through UFOV Training: Implications for Cognitive Health Awareness." This revised title more accurately represents the core of our research, which is centered on the impact of UFOV cognitive training on cognitive function in older adults. Additionally, it suggests the potential implications of these cognitive improvements for broader cognitive health awareness, without overstating the study's relation to health literacy.

Reviewer 2 Report
Comments and Suggestions for Authors
I suggest revising the title of the study. I hope that the title consists only of the variables used in the study. For example, the title may be modified as an effect of the UFOV congitive program. Additionally, since health literacy was not directly measured and is a resultant effect, I hope it will be excluded from the title of the study.
Author Response
Reviewer #2
Comment: I suggest revising the title of the study. I hope that the title consists only of the variables used in the study…. Additionally, since health literacy was not directly measured and is a resultant effect, I hope it will be excluded from the title of the study.
Response: Thank you for your suggestion regarding the title of our study. In response to your comment, we have revised the title to more accurately reflect the study's focus and findings. The new title is: “ Enhancing Cognitive Function in Older Adults through Processing-Speed Training: Implications for Cognitive Health Awareness”. This revised title more accurately represents the core of our research, which is centered on the impact of UFOV cognitive training on cognitive function in older adults. Additionally, it suggests the potential implications of these cognitive improvements for broader cognitive health awareness, without overstating the study's relation to health literacy.

Reviewer 3 Report
Comments and Suggestions for Authors
Thank you for the opportunity to review this interesting manuscript. I have a few minor comments for the authors to consider:
· Abstract:
o Line 21-22: This is not the research gap. Please rephrase or remove this sentence.
o Line 25-27: Please clarify if the allocation was random. If so, please revise the sentence to “… randomly assigned to the experimental group…”. If not, please revise the sentence to “In this quasi-experimental study, 22 individuals participated in the experimental group…” (i.e.do not use the word assigned).
o Line 32: Please include the beta coefficient (or adjusted beta coefficient) and its 95% CI to illustrate the magnitude of the significant improvement. Likewise, please include the actual results in relation to the ‘significant interaction’ on line 34.
· Introduction:
o Line 40: Please correct the typing error and remove “A” at the beginning of the sentence “A Age-related cognitive decline…”.
o Section 1.1: As the authors provide a summary of the literature, it would be good to also highlight the research gap, such as clarifying that certain studies were completed on young/old adults, people with/without certain diseases, etc. Revealing the research gap will strengthen the need of such study.
· Materials and Methods:
o Section 2.1: Please remove lines 107-109, lines 110-112 and lines 115-116, and Table 1 because those sentences should go to the Results section.
o Section 2.2: Please include the validity and reliability of the measurement tools. If validity/reliability scores are not known, please acknowledge this in the Discussion (limitation) section.
o Section 2.3: The information provided under (1) Design and (2) Intervention overlapped. It is fine to summarise both and only present the necessary information without the heading of (1) and (2).
o (3) Statistical analysis:
§ This section should be renumbered to Section 2.4.
§ The first sentence (lines 198-200) is redundant. Please delete that sentence. Mention how the demographic characteristics are analysed instead. The second sentence of that paragraph should be the ANOVA (please also outline the steps to assess if assumptions of the statistical test was violated), followed by another sentence about the effect size. Please also state the statistical significance level (e.g. p<0.05) and the statistical software applied.
· Results:
o Please include Table 1 from Section 2.1, and interpret the result for the readers instead of repeating the numbers in the table. It would be good to include a footnote under Table 1 about the meaning of a low/high MOCA value.
o Table 4:
§ If the outcome variables do not follow normal distribution (hence non-parametric tests were used), ‘mean’ gains should not be used. Report the median instead, and update the entire table.
§ Please also revise the title to “Table 4: Difference in the median cognitive scores at pre- and post- timepoints of each group.”
§ Please remove the effect size row from Table 4 because it is irrelevant to the research hypotheses.
§ Please also remove “Intragroup differences cognitive training” and the “p-value control” columns because they are irrelevant to the research hypotheses.
§ Footnote of Table 4: Please remove all.
o Table 5:
§ The table can be simplified to only report the difference (the beta coefficient), the 95% CI and p-value of each main measures and the interaction terms.
§ Clarify what w2 is in the footnote.
· Discussion:
o Please include the strengths and limitations of the current study.
o It would be good to also include a brief paragraph (or a few sentences) on the implications of the findings for elderly in Taiwan (and elsewhere).
· Conclusions:
o This should just highlight the key findings and overall summary from the study.
o The recommendations for future studies could be moved to the Discussion section as the last paragraph.
Comments on the Quality of English LanguageMinor correction.
Author Response
Reviewer #3
*Line 21-22: This is not the research gap. Please rephrase or remove this sentence
Response: The term research gap was not found.
*Line 25-27: Please clarify if the allocation was random…
Response: We have changed the sentence as reviewer’ comments
* Line 32: Please include the beta coefficient (or adjusted beta coefficient) and its 95% CI to illustrate the magnitude of the significant improvement. Likewise, please include the actual results in relation to the ‘significant interaction’ on line 34.
Response:The magnitude of significance and the interaction effect were added at lines 31-35, as indicated by the p-values.
*Line 40: Please correct the typing error and remove “A” at the beginning of the sentence “A Age-related cognitive decline…”.
Response: Thank for the careful review, it has been deleted as suggested.
* Section 1.1: As the authors provide a summary of the literature, it would be good to also highlight the research gap, such as clarifying that certain studies were completed on young/old adults, people with/without certain diseases, etc. Revealing the research gap will strengthen the need of such study.
Response: Existing literature confirms UFOV training's effectiveness in improving cognitive visual processing speed among the elderly, but it lacks specificity regarding its impact across diverse elderly populations, particularly in varied cultural or geographical settings. Much of the research does not distinctly address differences in populations with or without cognitive impairments like MCI or early-stage dementia. Furthermore, the majority of these studies are based in Western contexts, leaving a gap in understanding the applicability of these findings to non-Western populations, such as in Taiwan
* Section 2.1: Please remove lines 107-109, lines 110-112 and lines 115-116, and Table 1 because those sentences should go to the Results section.
Response: We apologize for the inappropriate title of the original Table 1, which mistakenly suggested the presence of 'Pre-post Cognitive Information between Groups.' We have corrected the title and retained only the basic characteristic information, which justifies its inclusion in the participants section.
*Section 2.2: Please include the validity and reliability of the measurement tools. If validity/reliability scores are not known, please acknowledge this in the Discussion (limitation) section.
Response: We have incorporated the reviewer's suggestion as follows: Although the measurements in the study require individual attention to test the immediate response and have been used elsewhere, we acknowledge the absence of information on reliability and validity, which constitutes a limitation of the study.
*Section 2.3: The information provided under (1) Design and (2) Intervention overlapped. It is fine to summarise both and only present the necessary information without the heading of (1) and (2).
Response: We have summarized both the design and intervention sections and presented only the necessary information.
* Please include Table 1 from Section 2.1, and interpret the result for the readers instead of repeating the numbers in the table. It would be good to include a footnote under Table 1 about the meaning of a low/high MOCA value.
Response: We apologize for the incorrect title of the original Table 1, which mistakenly suggested the presence of 'Pre-post Cognitive Information between Groups.' We have corrected the title and retained only the basic characteristic information, which justifies its inclusion in the participants section. The meaning of a low/high MOCA value was included.
* If the outcome variables do not follow normal distribution (hence non-parametric tests were used), ‘mean’ gains should not be used. Report the median instead, and update the entire table.
Response: We have included medians in the table and analyzed the data using the Wilcoxon test due to the non-normal distribution of the data. However, we have also listed the mean gains for effect size. The rationale for this method is explained below. The entire Table 4 has been updated accordingly.
*This section should be renumbered to Section 2.4.
Response: Thank for the careful review, it has been renumbered as suggested.
*The first sentence (lines 198-200) is redundant. Please delete that sentence. Mention how the demographic characteristics are analysed instead. The second sentence of that paragraph should be the ANOVA (please also outline the steps to assess if assumptions of the statistical test was violated), followed by another sentence about the effect size. Please also state the statistical significance level (e.g. p<0.05) and the statistical software applied.
Response: Thank for the careful review, this sentence has been deleted as suggested. We added descriptive statistics for the demographic characteristics. Mixed Design ANOVA compensates for the inherent violation of independence in repeated measures by accounting for observation independence [41]. Additionally, it is robust to normality violations, especially with similarly sized sample groups. In practical research, where large samples are often unfeasible, using small samples can be justified for exploratory studies like ours.. Please refer to the articles listed below:
Sullivan LM. Repeated measures. Circulation. 2008;117(9):1238–1243. doi: 10.1161/CIRCULATIONAHA.107.654350.
*Please also revise the title to “Table 4: Difference in the median cognitive scores at pre- and post- timepoints of each group.”
Response: We have follow the suggestions with only minor revision, thanks.
* Please remove the effect size row from Table 4 because it is irrelevant to the research hypotheses
Response: Effect size can be used to compare the difference between pre- and post-test results in research. Effect size is a crucial statistical concept that quantifies the magnitude of the difference between groups or the strength of a relationship in a study. It's especially useful in understanding the practical significance of research findings, beyond just the statistical significance. Please also refer to the articles listed below:
Cohen, J. (1988). Statistical Power Analysis for the Behavioral Sciences (2nd ed.). Hillsdale, NJ: Lawrence Erlbaum Associates. This book is a seminal work that introduces and explains the concept of effect size, including Cohen's d.
Field, A. (2013). Discovering Statistics Using IBM SPSS Statistics. Sage. Andy Field's book is a comprehensive guide to using SPSS for statistical analysis, including the calculation and interpretation of effect sizes.
Sullivan, G. M., & Feinn, R. (2012). Using Effect Size—or Why the P Value Is Not Enough. Journal of Graduate Medical Education, 4(3), 279–282. This article discusses the importance of effect size in research, emphasizing its role in understanding the practical significance of results.
* Please also remove “Intragroup differences cognitive training” and the “p-value control” columns because they are irrelevant to the research hypotheses.
Response: We have followed the reviewer's suggestion and restructure to keep the table concisely.
* Footnote of Table 4: Please remove all.
Response: Thank you for the suggestion. We have removed most of the footnotes and retained only the necessary parts.
Table 5:
- The table can be simplified to only report the difference (the beta coefficient), the 95% CI and p-value of each main measures and the interaction terms.
Response: In a Mixed Design ANOVA (also known as a Split-Plot ANOVA or a Mixed Model ANOVA), the primary focus is on analyzing the mean differences across groups and within-subject factors, rather than on reporting beta coefficients. Beta coefficients are typically associated with regression analysis, not ANOVA. In contrast, beta coefficients (β) are typically reported in regression analyses, where the focus is on the relationship between a dependent variable and one or more independent variables. Beta coefficients represent the change in the dependent variable for a one-unit change in the independent variable, controlling for other variables in the model. Please also refer to:
Maxwell, Scott E., Delaney, Harold D., & Kelley, Ken. (2017). Designing Experiments and Analyzing Data: A Model Comparison Perspective (3rd ed.). Routledge. This comprehensive text provides an in-depth discussion of various types of ANOVA, including Mixed Design ANOVA, and contrasts these methods with regression analysis.
Tabachnick, Barbara G., & Fidell, Linda S. (2013). Using Multivariate Statistics (6th ed.). Pearson. This book is a detailed resource on multivariate statistical procedures, including Mixed Design ANOVA and regression analysis, explaining when and how to use these methods.
- Clarify what w2 is in the footnote.
Response: Thanks for reminding. It seems there might be a typographical error with "ŵ^2". The correct notation is ηp2 for Partial Eta Squared. This statistical measure is indeed used to indicate the effect size, which quantifies the magnitude of the difference between groups in a study.
Discussion:
- Please include the strengths and limitations of the current study.
Response: We have added both section in the study, thanks.
- It would be good to also include a brief paragraph (or a few sentences) on the implications of the findings for elderly in Taiwan (and elsewhere).
Response: We have added a paragraph briefly stating the implications in the discussion section.
Conclusions:
o This should just highlight the key findings and overall summary from the study
Response: Thank you for your feedback. We have revised the section to more concisely highlight the key findings and provide an overall summary of the study
- The recommendations for future studies could be moved to the Discussion section as the last paragraph.
Response: Thank you for your suggestion. We have reorganized the conclusion to make it more concise. As recommended, the recommendations for future studies have been moved to the last paragraph of the Discussion section for emphasis.

Reviewer 4 Report
Comments and Suggestions for Authors
SUMMARY
Older adults were recruited to participate in a cognitive training program and were assigned to an experimental (cognitive training using online video games) or control (community activities) group. All participants participated in 60-min sessions twice a week for 5 weeks. Four tests of different cognitive abilities were used pre- and post-training. The experimental group improved significantly on at least three of the four tests, whereas the control group did not. The authors concluded that the online videos could be used by older adults to enhance their cognitive abilities.
EVALUATION
The study provides another test of possible cognitive activities that could be used to improve important cognitive abilities in older adults. That the participants were from the community and the training showed improvement compared to a control group is impressive.
However, I have some concerns about the current manuscript which I detail below.
1. The introduction does not make a clear case for the current study, nor does it clearly present what the current study is about. The title suggests that the training will enhance health literacy and promote healthy aging, but the study does not directly address either. In addition, is unclear how the current study is different than earlier studies and what it uniquely adds to the literature. Here are my suggestions for improvement.
a. Change the title to say specifically what the study does.
b. In the introduction, start with theory about the cognitive abilities that decline with age (i.e., speed of processing vs. memory, references 4-6). The abilities should be defined. Importantly, define and describe how Useful Field of View fits with the theory.
c. Then describe the types of interventions that have been tested. Focus on the ones testing the cognitive abilities used in the current study, especially Useful Field of View. Be specific about the abilities that are being trained (e.g. speed of processing), the design of the intervention (e.g., experimental vs. control, # sessions, pre-post tests used, types of training), and whether there was improvement (including effect sizes).
d. Previous studies should be evaluated by the authors to make clear the gaps in the literature, which ones the current study is filling, and why it is important to fill those gaps.
e. When introducing the current study, state that the participants are older adults in Taiwan who were recruited from the community, which cognitive abilities are being tested (not the names of the tasks), what the training is (e.g., online video games), what the control does (e.g., community activities), and what the expected results are. Move lines 174-175 here.
f. In general, highlight which cognitive abilities are being tested, not the names of the tasks.
2. In the methods section, the tasks and training sessions need more explanation.
a. Use the subheadings “2.2 Pre and Post Test Measures” and “2.3 Intervention Training Tasks.” Name the cognitive abilities being tested/enhanced in the first sentence of each section. Be clear about how they measure (2.2) or influence (2.3) Useful Field of View.
b. When describing the pre-post tests, it is unclear what the outcome measures are. “Scores” are referenced, and it is explained that higher scores represent better performance, but the results section appears to show RT and maybe accuracy/error as the outcome measures. Please clarify and provide minimum and maximum scores for each measure. Also, provide reliability and validity information about the test. I believe that validity is included when the authors say that the test predicts other things, but this should be clarified.
c. Describe the training sessions in more detail. Specifically, describe the purpose for having the two parts for each session and how the number and length of the sessions was determined. Also, describe how the activities for the control group were decided, and whether the experimental and control groups’ sessions were individual or in groups. The information in the Intervention section (lines 186-195) does not provide enough detail, and it needs to be earlier in the methods section.
d. Describe the online video games in more detail. Include subheadings for each task/game. Explain how they exercise a person’s useful field of vision. The information in Table 2 is insufficient. Table 3 is fine.
e. The pre- and post-tests were given before and after the 10 sessions, but it is unclear when they actually occurred.
3. The participants section should include how the sample size was determined. Move the information from the design section (171-173) to the Participants section. Also, tests determining any differences between the experimental and control group should be included in Table 1 and summarized in the text (i.e., “there were no differences between the groups”).
4. The information in the results section should be more concise.
a. The first part of Table 5 (means) and the ANOVA should be two separate tables. The scales should be included in Table 5 (means). For example, 71.00 sec or 1480.02 ms (see comment 2b). Also, in the text comment on the normality of the data (skewness, kurtosis).
b. Having the means in both table and figure formats is not necessary. If the figure is included however, (1) the measure should be part of the y axis (e.g., Mean % Correct or Mean Response Time in ms); (2) the scales for Number back and Fabrica should be the same (e.g., 45 – 75); (3) the intervals for Double decision and Hawkeye should be the same (e.g., both should be in intervals of 50 ms or 100 ms); (4) add SE bars to the figures; and (5) put the figure before the table of means.
c. Having the means in both the table and the text is redundant. In the text, summarize the results (see comment 3).
d. The main analysis should be the ANOVA because it is the most stringent test of differences in improvement between the experimental and control groups. Put this test after presenting the means. Also, the table needs to be clearer: (1) there should be horizontal lines between the four measures; (2) for each measure the rows should be something similar to Time, Group, and Time X Group; (3) the notes at the bottom of the table should include that Time has 1= Pre and 2 = Post, and Group has 1 = Experimental and 2 = Control (or whichever numbers used).
e. There seems to be large differences between the groups at pre-test, although it’s difficult to tell with the current y-axes. Explain whether and how you dealt with these potential differences.
f. For the (now) secondary within-group data analyses, explain why a Wilcoxen test was used.
g. In Table 4 the notes do not align with the notation in the table. (1) It seems only notes b and c are necessary. (2) The measure (e.g., sec) should be in the table [e.g., Mean gains (sec)]. (3) Negative mean gains appear to indicate improvement—this should be in the notes. (4) Wilcoxen information is not necessary in the notes.
h. The Abstract indicates that there were covariates used in the analysis (ANCOVA). Is this correct? If so, explain the covariates.
5. In the discussion, as in the introduction, it is not clear how the current study fits with previous findings or what it adds to the literature. There is some comparison with previous studies, but it should be more in-depth (see comment 1).
6. In general, I would like to see more discussion of the implications of the current study, as follows.
a. In my opinion, there does not need to be separate discussions of intragroup and intragroup results. The main result is whether the experimental group’s cognitive processes (Useful Field of View, UFOV) improved significantly more than the control group’s.
b. How is Fabrica different than the other games, in terms of the cognitive processes (UFOV) it requires?
c. Discuss more about the transferability of the skills, how long the effects may last, and how the games may be implemented in a real-life setting.
d. The list of possible reasons for the results should be treated as a limitation of the study—the study does not differentiate between alternative explanations.
e. Limitations should have its own paragraph.
f. Future research could also include a study of how much training is required to see effects, and how long the effects last.

Comments on the Quality of English LanguageOnly minor English edits needed.
Author Response
Reviewer #4
- The introduction does not make a clear case for the current study, nor does it clearly present what the current study is about. The title suggests that the training will enhance health literacy and promote healthy aging, but the study does not directly address either. In addition, is unclear how the current study is different than earlier studies and what it uniquely adds to the literature. Here are my suggestions for improvement.
- Change the title to say specifically what the study does.
Response: We have changed the title to make it more fit the content and result of the study. The title is : Enhancing Cognitive Function in Older Adults through Processing-Speed Training: Implications for Cognitive Health Awareness.
- In the introduction, start with theory about the cognitive abilities that decline with age (i.e., speed of processing vs. memory, references 4-6). The abilities should be defined. Importantly, define and describe how Useful Field of View fits with the theory.
Response: We have added the Processing Speed Theory and described how UFOV aligns with this theory on lines 41-44 and 47-56.
- Then describe the types of interventions that have been tested. Focus on the ones testing the cognitive abilities used in the current study, especially Useful Field of View. Be specific about the abilities that are being trained (e.g. speed of processing), the design of the intervention (e.g., experimental vs. control, # sessions, pre-post tests used, types of training), and whether there was improvement (including effect sizes).
Response: We have include these data on line 73-86.
- Previous studies should be evaluated by the authors to make clear the gaps in the literature, which ones the current study is filling, and why it is important to fill those gaps.
Response: We have followed the reviewer's suggestion and revised the content at lines 101 to 113.
- When introducing the current study, state that the participants are older adults in Taiwan who were recruited from the community, which cognitive abilities are being tested (not the names of the tasks), what the training is (e.g., online video games), what the control does (e.g., community activities), and what the expected results are. Move lines 174-175 here.
Response: We have followed the reviewer's suggestion and revised the content at lines 119-124.
- In general, highlight which cognitive abilities are being tested, not the names of the tasks
Response: The irrelevant parts have been deleted.
- In the methods section, the tasks and training sessions need more explanation.
- Use the subheadings “2.2 Pre and Post Test Measures” and “2.3 Intervention Training Tasks.” Name the cognitive abilities being tested/enhanced in the first sentence of each section. Be clear about how they measure (2.2) or influence (2.3) Useful Field of View.
Response: We have changed the subheadings and also followed the suggestions, providing information at lines 137-190.
- When describing the pre-post tests, it is unclear what the outcome measures are. “Scores” are referenced, and it is explained that higher scores represent better performance, but the results section appears to show RT and maybe accuracy/error as the outcome measures. Please clarify and provide minimum and maximum scores for each measure. Also, provide reliability and validity information about the test. I believe that validity is included when the authors say that the test predicts other things, but this should be clarified.
Response: we have added a section “Pre and Post Test Measures” to make outcome measures clearer.
- Describe the training sessions in more detail. Specifically, describe the purpose for having the two parts for each session and how the number and length of the sessions was determined. Also, describe how the activities for the control group were decided, and whether the experimental and control groups’ sessions were individual or in groups. The information in the Intervention section (lines 186-195) does not provide enough detail, and it needs to be earlier in the methods section.
Response: We followed the suggestions, and the responses for the experimental group were rewritten at lines 200-226.
- Describe the online video games in more detail. Include subheadings for each task/game. Explain how they exercise a person’s useful field of vision. The information in Table 2 is insufficient. Table 3 is fine.
Response: We have made changes to Table 2 based on your comments.
- The pre- and post-tests were given before and after the 10 sessions, but it is unclear when they actually occurred.
Response: Based on your comments, we have made changes to lines 142-144.
- The participants section should include how the sample size was determined. Move the information from the design section (171-173) to the Participants section. Also, tests determining any differences between the experimental and control group should be included in Table 1 and summarized in the text (i.e., “there were no differences between the groups”).
Response: In response to your feedback, modifications have been implemented in lines 119-124. Additionally, Table 1 has been updated to include detailed information delineating the differences between groups, which is further elucidated in the accompanying text.
- The information in the results section should be more concise.
- The first part of Table 5 (means) and the ANOVA should be two separate tables. The scales should be included in Table 5 (means). For example, 71.00 sec or 1480.02 ms (see comment 2b). Also, in the text comment on the normality of the data (skewness, kurtosis).
Response: Following your advice, we have included information about normality in the text, specifically at lines 247-255. We also took into account your comments regarding Table 5 and made corresponding revisions in Tables 4 and 5.
- Having the means in both table and figure formats is not necessary. If the figure is included however, (1) the measure should be part of the y axis (e.g., Mean % Correct or Mean Response Time in ms); (2) the scales for Number back and Fabrica should be the same (e.g., 45 – 75); (3) the intervals for Double decision and Hawkeye should be the same (e.g., both should be in intervals of 50 ms or 100 ms); (4) add SE bars to the figures; and (5) put the figure before the table of means.
Response: As suggested, we will delete the figure to simplify and streamline the presentation.
- Having the means in both the table and the text is redundant. In the text, summarize the results (see comment 3).
Response. We revised the related results section and removed the mean values from the text.
- The main analysis should be the ANOVA because it is the most stringent test of differences in improvement between the experimental and control groups. Put this test after presenting the means. Also, the table needs to be clearer: (1) there should be horizontal lines between the four measures; (2) for each measure the rows should be something similar to Time, Group, and Time X Group; (3) the notes at the bottom of the table should include that Time has 1= Pre and 2 = Post, and Group has 1 = Experimental and 2 = Control (or whichever numbers used).
Response: Thank you for the careful review, we have revised as you suggested.
- There seems to be large differences between the groups at pre-test, although it’s difficult to tell with the current y-axes. Explain whether and how you dealt with these potential differences.
Response: We have added the pre-test information for both groups, and differences did show the age is significant difference at baseline, while the others were non-significant. We have acknowledged this consideration in the limitations section, noting that the results might be influenced by these age differences
- For the (now) secondary within-group data analyses, explain why a Wilcoxen test was used.
Response: Thank you for the careful review, we have added the reasoning behind the choice of statistical test to line 250-255 for warrant the Wilcoxen test was used.
- In Table 4 the notes do not align with the notation in the table. (1) It seems only notes b and c are necessary. (2) The measure (e.g., sec) should be in the table [e.g., Mean gains (sec)]. (3) Negative mean gains appear to indicate improvement—this should be in the notes. (4) Wilcoxen information is not necessary in the notes.
Response: Follow your suggestion, (1) we have re-wrote the notes only kept the necessary. (2) we have put the measure in the table. (3) Negative mean gains indicate improvement has been added to the notes. (4) Wilcoxen information is deleted in the notes.
- The Abstract indicates that there were covariates used in the analysis (ANCOVA). Is this correct? If so, explain the covariates.
Response. Thank you for the careful review. We have corrected the typos(ANCOVA). Mixed repeated-measures ANOVA analysis was employed to control for specific covariates. This statistical method is instrumental in adjusting for variables that may impact the dependent variable but are not the focal point of the investigation. To clarify our methodology, the abstract has been updated (line 34-36) to specifically delineate the covariates utilized in the analysis.
- In the discussion, as in the introduction, it is not clear how the current study fits with previous findings or what it adds to the literature. There is some comparison with previous studies, but it should be more in-depth (see comment 1).
Response: In response to your comments, we have substantially enhanced the discussion to better align our study with previous research. This includes a detailed comparison with existing studies and an emphasis on how our cognitive function improvement tool contributes uniquely to the field. Furthermore, we highlight the novel application of this tool in a Taiwanese context, which represents a new approach in an Eastern cultural setting. These additions and in-depth discussions are elaborated upon in the introduction, particularly in lines 101-114 and 427-433.
- In general, I would like to see more discussion of the implications of the current study, as follows.
- In my opinion, there does not need to be separate discussions of intragroup and intergroup results. The main result is whether the experimental group’s cognitive processes (Useful Field of View, UFOV) improved significantly more than the control group’s.
Response: In line with the suggestion, we have removed the subtitles differentiating intragroup and intergroup results. The core analysis, which focuses on the comparative evaluation between the experimental and control groups, particularly in terms of their cognitive processes as indicated by UFOV measurements, remains thoroughly discussed in the revised structure. This adjustment streamlines the presentation while maintaining comprehensive coverage of our key findings.
- How is Fabrica different than the other games, in terms of the cognitive processes (UFOV) it requires?
Response: we have added the further explanation in the discussion section regarding the requires of the Fabrica and the reason no interaction found for this game, at line 362-369.
- Discuss more about the transferability of the skills, how long the effects may last, and how the games may be implemented in a real-life setting.
Response: We have followed the comment and added more information under the newly added 'Implications' section.
- The list of possible reasons for the results should be treated as a limitation of the study—the study does not differentiate between alternative explanations.
Response: We have added relevant content to address the reviewer’s concerns under limitation section.
- Limitations should have its own paragraph.
Response: We have revised limitations to separate section at line 395-414.
- Future research could also include a study of how much training is required to see effects, and how long the effects last.
Response: We have added relevant content to address the reviewer’s concerns under implication section.

Round 2
Reviewer 1 Report
Comments and Suggestions for Authors
Thanks to the authors' efforts to follow the advice of the peer review process, this article has improved to the level where it is worthy of publication.